# Aluminum Thin Film Nanostructure Traces in Pediatric EEG Net for MRI and CT Artifact Reduction

**DOI:** 10.3390/s23073633

**Published:** 2023-03-31

**Authors:** Hongbae Jeong, Georgios Ntolkeras, Tracy Warbrick, Manfred Jaschke, Rajiv Gupta, Michael H. Lev, Jurriaan M. Peters, Patricia Ellen Grant, Giorgio Bonmassar

**Affiliations:** 1AA. Martinos Center, Massachusetts General Hospital, Harvard Medical School, Charlestown, MA 02129, USA; 2Department of Newborn Medicine, Fetal-Neonatal Neuroimaging and Developmental Science Center, Boston Children’s Hospital, Boston, MA 02115, USA; 3Department of Pediatrics, Baystate Medical Center, University of Massachusetts Medical School, Springfield, MA 01605, USA; 4Brain Products GmbH, 82205 Gilching, Germany; 5Department of Radiology, Massachusetts General Hospital, Harvard Medical School, Boston, MA 02114, USA; 6Department of Neurology, Boston Children’s Hospital, Harvard Medical School, Boston, MA 02115, USA

**Keywords:** antenna effect, B_1_ artifact, RF-induced currents, streak artifact

## Abstract

Magnetic resonance imaging (MRI) and continuous electroencephalogram (EEG) monitoring are essential in the clinical management of neonatal seizures. EEG electrodes, however, can significantly degrade the image quality of both MRI and CT due to substantial metallic artifacts and distortions. Thus, we developed a novel thin film trace EEG net (“NeoNet”) for improved MRI and CT image quality without compromising the EEG signal quality. The aluminum thin film traces were fabricated with an ultra-high-aspect ratio (up to 17,000:1, with dimensions 30 nm × 50.8 cm × 100 µm), resulting in a low density for reducing CT artifacts and a low conductivity for reducing MRI artifacts. We also used numerical simulation to investigate the effects of EEG nets on the B_1_ transmit field distortion in 3 T MRI. Specifically, the simulations predicted a 65% and 138% B_1_ transmit field distortion higher for the commercially available copper-based EEG net (“CuNet”, with and without current limiting resistors, respectively) than with NeoNet. Additionally, two board-certified neuroradiologists, blinded to the presence or absence of NeoNet, compared the image quality of MRI images obtained in an adult and two children with and without the NeoNet device and found no significant difference in the degree of artifact or image distortion. Additionally, the use of NeoNet did not cause either: (i) CT scan artifacts or (ii) impact the quality of EEG recording. Finally, MRI safety testing confirmed a maximum temperature rise associated with the NeoNet device in a child head-phantom to be 0.84 °C after 30 min of high-power scanning, which is within the acceptance criteria for the temperature for 1 h of normal operating mode scanning as per the FDA guidelines. Therefore, the proposed NeoNet device has the potential to allow for concurrent EEG acquisition and MRI or CT scanning without significant image artifacts, facilitating clinical care and EEG/fMRI pediatric research.

## 1. Introduction

Central nervous system disorders in neonates may present as encephalopathies and are often accompanied by seizures. Acute symptomatic seizures in neonates may stem from perinatal hypoxic-ischemic injury or stroke, but epilepsy in neonates can also stem from genetic-metabolic etiologies or brain structural abnormalities. Magnetic resonance imaging (MRI) plays a crucial role in diagnosing and understanding the cause of neonatal seizures. However, neonatal MRI evaluation only assesses the structure, and not the function [1]. In such circumstances, continuous video EEG provides essential information about epileptiform abnormalities and seizures (frequency, distribution, and location), and the overall brain function is reflected in the background pattern. The latter evolves with gestational age and has age-specific norms describing typical graphic elements and natural background patterns in the awake and sleep states. In critically ill neonates, infants, and young children in the ICU, EEG monitoring is the only widely accepted clinical tool that provides a continuous and direct window to brain function [2]. EEG is critical in the diagnosis of seizures, for monitoring the recovery from hypoxic-ischemic events, and in the assessment of brain maturity. In all of these situations, high-density EEG (HD-EEG) is preferred as it provides the best signal-to-noise ratio (SNR), dynamic range, and sampling rate compared to the older 10–20 EEG. Furthermore, HD-EEG has enhanced artifact removal and can automatically detect seizures [3] (see also Appendix A). Typically, however, EEG nets must be removed prior to MRI and CT imaging, as they can pose a safety hazard in the MRI environment and can severely impact the image quality. The removal and re-application process of the EEG net is time-consuming and labor-intensive and can result in long periods where the EEG cannot be monitored, potentially impacting patient care and limiting the ability of the health care teams to make well-informed decisions.

This paper presents a novel high-density EEG net (i.e., NeoNet see Figure 1) using an aluminum thin film nanostructure for cloaking by computerized tomography (CT) and MRI. The NeoNet is a step forward compared to the state-of-the-art polymer thick film nets [4] by providing more accurate control of the trace electrical properties. Here, we report a scalable approach to produce micro and nanoscale structures (Figure 2) of aluminum EEG traces with an *ultra-high-aspect ratio* (up to 17,000:1, with dimensions 30 nm × 50.8 cm × 100 µm). Thus, the real challenge is to construct long, flexible, and narrow aluminum thin film traces for the NeoNet. To the best of our knowledge, this has not been reported yet. The major strengths of the NeoNet lie in the ease of application of the net compared to single electrodes in fragile patients such as neonates and children, and its compatibility with MRI environments, not needing to remove electrodes prior to the recording and re-apply them after the exam. Finally, other potential applications of the NeoNet include electrical impedance spectroscopy (EIS) in children [3].

The aluminum thin films and alloys comprise most of the interconnections used in semiconductor chips [5]. Thin film is an exciting technology for fabricating nanoscale electric traces and circuits. Aluminum thin films have been shown to be advantageous in nanoscale stacked transistor fabrication at room temperature [6], as in the case of the construction of a neonatal EEG net (NeoNet), which enables working with polymer substrates or papers for displays [7], avoids the use of rare-earth elements, and produces no high-temperatures toxic by-products [8]. Since EEG leads may act as an antenna and may capture RF waveforms induced by the MRI [9], may generate B_1_ artifacts, and may introduce safety issues [10,11], the ideal stripline design for MRI compatibility is the one that minimizes the current flow at RF, which in our case was 128 MHz (3 Tesla MRI), while maintaining the SNR intact at low frequency (<1 kHz). Ideally, materials with high $\mu_{r}$ would exhibit such behavior. However, metals with high $\mu_{r}$ will also produce significant MRI susceptibility (i.e., B_0_) artifacts, thus traces with low conductivity that make traces of 10 kΩ (10 kΩ resistors are inserted between leads and EEG electrodes in commercial EEG/MRI sets [10]) of resistance or more are ideal. Aluminum is again a good candidate since it has a high bulk resistivity (i.e., 2.65 10^−8^ Ω·m [12]) among all of the non-alloy, non-brittle, biocompatible [13], and non-magnetic metals (e.g., chromium is antiferromagnetic at room temperature). Aluminum is also the least dense (i.e., 2.65 g/cc) non-brittle elemental metal (e.g., titanium is almost twice as dense as Al at 4.5 g/cc), which confers Al with a low CT artifact material property [14]. Finally, mechanical stress can occur if the traces are potentially pulled, especially in pediatric applications, and of course, we want the traces to remain intact both electrically and mechanically. The tensile properties of free-standing electron-beam-deposited aluminum thin films have been studied [15], where a high σ_y_ ductility in terms of large % elongation and ultimate tensile strength (UTS) was observed mainly due to the fine grain sizes. The Young’s modulus (measuring the deformation when unloading-reloading), albeit less important, was lower than half of the reported value for pure aluminum in the bulk form.

In this study, we built the NeoNet, conducted CT image quality testing, MRI safety and image quality studies, and a series of electromagnetic simulations to estimate the MRI quality of a 29-month-old child wearing the NeoNet using a finite-difference time-domain (FDTD) method. The results of the thin film-based EEG trace (NeoNet) were compared with NoNet, CuNet (without resistors), and CuNet (with ideal current limiting resistors) in terms of the B_1_ transmit field distortion. The paper is divided into a theoretical part, which illustrates the physical principles of CT and MRI cloaking, and a methods and results section that explains how the fabrication, measurements, and numerical simulations were performed and the resulting data. Finally, a discussion regarding the fabrication process, safety, numerical simulations, MRI, and CT measurements is presented as well as the summary conclusions.

## 2. Theory

A CT scan reconstructs an image of the human body by combining the photon incident counts produced by X-ray generators at different angles and by taking the negative logarithm, yields samples of the Radon transform of the linear attenuation map. However, such photons are scattered and/or attenuated by the presence of metals inside or outside the body such as EEG leads, producing hardening or streak artifacts in regions potentially of interest (e.g., brain lesions). Metal artifacts may decrease lesion identification for radiation diagnosis and reduce the accuracy of radiotherapy’s target delineation and dose calculation. In order to reduce the presence of hardening artifacts, we adopted our twin approach of using a very thin metal film (i.e., 30 nm) and selected aluminum deposition, given its very low mass density (see Table 1). Furthermore, aluminum was chosen as it is flexible, non-toxic, and, most of all, it is non-magnetic, thus will produce only minimal magnetic susceptibility artifacts [16]. Thin aluminum film traces were selected as we will show that they are also capable of producing real and imaginary losses to reduce the *antenna effect* [10] and generate the desired MRI cloaking effect of the EEG traces, thanks to the relatively low conductivity.

For simplicity, we initially studied the geometry (Appendix A) of a birdcage coil and a thin conductive rectangular trace with length *l*, width *w*, and thickness *t*, oriented along the *z*-axis at a distance *r* from the center and at an angle φ from the *x*-axis. The MRI coil produces a rotating radiofrequency B_1_ time-varying field, which induces in the trace a current along the trace that may introduce artifacts in the B_1_ field according to the following one-dimensional wave equation:(1)$$\;\frac{\partial^{2}}{\partial z^{2}}A_{z}\left(z\right)=\left(\varepsilon_{r}-\frac{j}{\omega \varepsilon_{0}\rho }\right)k^{2}A_{z}\left(z\right)$$
where ρ is the resistivity [Ω·m] and $\varepsilon_{r}$ is the electrical permittivity of the bulk conductive trace, $k=\omega \sqrt{\varepsilon_{r}\mu_{0}}$ is the wave number of free space, and $A_{z}\left(z\right)$ is the magnetic vector potential with the solution of Equation (1) [17]:(2)$$A_{z}\left(z\right)=\frac{\mu_{0}}{4\pi }\int_{-l/2}^{l/2}J_{z}\left(z\right)\frac{e^{-jk\sqrt{{\left(z-z^{\prime }\right)}^{2}+tw}}}{\sqrt{{\left(z-z^{\prime }\right)}^{2}+tw}}dz^{\prime }$$

The current density induced by the MRI coil is:(3)$$J_{z}\left(z\right)=-j\omega \left(\frac{1}{\rho }+j\omega \varepsilon_{0}\varepsilon_{r}\right)A_{z1}\left(z\right)=-j\omega \sigma^{*}A_{z1}\left(z\right)$$

For a birdcage coil with a radius *R*, the *z*-axis magnetic vector potential generated is given by the following Fourier series [18]:(4)$$A_{z1}\left(z\right)=\underset{m}{\sum }\frac{\mu_{0}S_{m}\left(z\right)}{2\left|m\right|}{\left(\frac{r}{R}\right)}^{\left|m\right|}e^{jm\varphi }\;\;$$
where $S_{m}$ are the surface currents along the birdcage coil. Therefore, the magnetic vector potential is:(5)$$A_{z}\left(z\right)=\frac{-j\omega \sigma^{*}\mu_{0}^{2}}{8\pi }\underset{m}{\sum }\frac{e^{jm\;}}{\left|m\right|}{\left(\frac{z}{h}\right)}^{\left|m\right|}\int_{-l/2}^{l/2}S_{m}\left(z\right)\frac{e^{-jk\sqrt{{\left(z-z^{\prime }\right)}^{2}+tw}}}{\sqrt{{\left(z-z^{\prime }\right)}^{2}+tw}}dz^{\prime }=-\frac{j\omega \sigma^{*}\mu_{0}^{2}}{8\pi }\Gamma_{z}$$

Thus, the $B_{1t}$ artifact induced by the trace is:(6)$$B_{1t}=-\frac{j\omega \sigma^{*}\mu_{0}^{2}}{8\pi }\nabla \times \Gamma_{z}$$

This is reduced by reducing $\sigma^{*}$, or equivalently by reducing the relative permittivity (however $\varepsilon_{r}=1$ in metals) and increasing the trace resistivity, which can be achieved by selecting metals with higher resistivity. The total trace resistance increases linearly with the metal resistivity:(7)$$R_{t}=\rho \frac{l}{wt}$$

However, in order for $R_{t}\geq 10\;k\Omega$, resistors were added at the end of the trace that are commonly used by the industry for safety [19], whereas the aluminum traces required a nanoscale thickness to avoid the need for additional resistors. Several studies have shown that the resistivity of aluminum conductors may increase just by reducing one dimension to the nanoscale [20,21,22], which is a desirable property since too thin a width in the traces may have manufacturing and reliability issues. The hypothesis behind this physical property was first proposed in 1938 by Fuchs [23], and this effect occurs when the nanoscale thickness of the metal is shorter than the mean free path of the conduction electrons, resulting in collisions with the boundaries of the film. The numerical model that predicted the thin film resistivity was subsequently improved [24,25] to correctly describe the resistivity changes with thickness in the case of different film morphologies (e.g., mono, polycrystalline films, etc.), and thus various types of electron scattering (e.g., background, grain boundaries, and surface scattering).

According to the combined Fuchs–Sondheimer [23] and Mayadas–Shatzkes [25] conduction model (MS-FS [26]), the ratio between the bulk and thin film resistivities is:(8)$$\frac{\rho_{0}}{\rho }=1-\frac{3}{2}\alpha +3\alpha^{2}-3\alpha^{3}Log\left(1+\frac{1}{\alpha }\right)-f\left(k\right)$$
where $\rho_{0}=2.65\cdot 10^{-8}$ [Ω · m] (i.e., bulk aluminum resistivity [27]) and:(9)$$f\left(k\right)=\frac{6\left(1-p\right)}{\pi k}\int_{0}^{\pi /2}\int_{1}^{\infty }\left(\frac{1}{y^{3}}-\frac{1}{y^{5}}\right)\frac{cos^{2}\vartheta \left(1-e^{-ktG\left(y,\vartheta \right)}\right)}{G{\left(y,\vartheta \right)}^{2}\left(1-pe^{-ktG\left(y,\vartheta \right)}\right)}d\vartheta dy$$
where p=0.510 is the fraction of electrons specularly scattered at the external surfaces [22,28], *k* is the film thickness *t* divided by the bulk electron mean free path $l_{0}$, and
(10)$$\left(y,\vartheta \right)=1+\frac{\alpha }{{\left(1-1/y^{2}\right)}^{2}cos\left(\vartheta \right)}$$
where $\alpha =\frac{l_{0}r}{a_{g}\left(1-r\right)},$ and $a_{g}$ is the crystalline size and *r* is the grain boundary reflection coefficient. The MS-FS model can be simplified as [29] (see Appendix A):(11)$$\frac{\rho_{0}}{\rho }≅1-\frac{3}{2}\alpha +3\alpha^{2}-3\alpha^{3}Log\left(1+\frac{1}{\alpha }\right)-\frac{3\left(1-p\right)}{8k}$$

According to our measurements *R* ≅ 2.5 kΩ when *l* = 0.463 [m] and *w* = $10^{-4}$ [Ω] (see the SEM in Appendix A). Thus, the minimum thickness based on the bulk resistivity of Al and Equation (7) was *t* ≅ 50 nm [21] (i.e., almost twice as thick compared to the value t = 30 nm reported in the datasheet of the aluminum metalized polymer film utilized, see Materials and Methods), while Equation (10) predicted a t≅1 μm based on k≅1, and α=2.5 [26]. In order to reach the desired $R_{t}$ we had to design traces with l ≅ 2.5 [m], but given our photolithography panel size limit, we deigned traces with N = 5 loops (Appendix A). The temperature-dependent overall resistance is:(12)$$R_{t}=\rho \frac{\left(N+1\right)\;l\;}{wt}\left[1+\alpha \cdot \left(T-20{}^{\circ}\right)\right]$$
where *T* is the temperature [°C] and *α* is the temperature coefficient [Ω/°C] (see Table 1).

## 3. Materials and Methods

This study was conducted in accordance with the Declaration of Helsinki and approved by the Institutional Boards of Massachusetts General Hospital (MGH) and Boston Children (BCH) Hospital involving human studies. Three experiments were performed, one at MGH and two at BCH, to evaluate the MRI quality in adults and the MRI and EEG quality, respectively. The adult was a 25-year-old female healthy subject with a small head size that was studied at 3 Tesla MRI (Subject 1), a five-year-old female with left-side parietal atrophy (Subject 2), and a five-year-old male with left-side tuberous malformations (Subject 3). The EEG was performed on a 5-year-old left-handed male with refractory focal seizures, subtle left frontotemporal focal cortical dysplasia on MR imaging, and complex left frontotemporal spikes (Subject 4). Furthermore, to show MRI images with EEG in the worst-case scenarios, we searched the Picture Archiving and Communicating System (PACS) system at MGH and found images of three subjects for MRI and CT.

### 3.1. NeoNet Construction

Sheets of 540 mm × 540 mm, 50 µm thick polyethylene terephthalate polyester (PETP) film already vacuum coated on one side with 30 nm aluminum were used (ES301955, GoodFellow, Coraopolis, PA, USA). The traces were composed of seven process fabrication steps (Figure 2): The traces were manufactured (TechEtch Inc., Plymouth, MA, USA) using photolithography, etching, pressure-mounting polyimide dielectric films, and laser cutting (Figure 2a,b). Since the electrodes were supposed to be in contact with the KCl solution and disinfectant solution that abraded the 30 nm thin aluminum, the electrodes were coated with a thin 300 nm layer of gold, with a 10 nm layer of chromium to improve adhesion (Figure 2c–e). The NeoNet electrodes were built based on a silver/silver chloride coating of thin (25 µm) pure silver sheets by a leading manufacturer of disposable silver/silver chloride coated electrode sensors. The process consisted of coating a silver foil, 0.025 mm (0.001″) thick, annealed, 99.95% (12190, Alfa Aesar, Tewksbury, MA, USA) with silver chloride (Ag/AgCl) with a chemical process. Ag/AgCl foils were cut into a disk of ¼″ diameter and glued the disks to the aluminum traces using a silver conductive epoxy adhesive (MG Chemicals, ON, Canada) and embedded in a custom-made holder (Brain Products, Germany) in contact with a sponge (Figure 2f,g and Appendix A).

### 3.2. MRI Safety Recordings

The RF safety of the NeoNet was tested in a 3 T MRI (Prisma, Siemens Healthineers) using a 29-month-old head-sized agar phantom. The target dielectric properties were chosen to match the averaged pediatric brain and skin properties at 128 MHz (ε_r_ = 74.95, σ = 0.64 (S/m) [30] (Appendix A). The MRI dielectric phantom recipe generator [31] was used to estimate the recipe for the target dielectric properties. The two test samples were obtained by mixing 1 L of distilled water (DI) with NaCl (purity 98%, Sigma-Aldrich Cop, MO), Sugar, and edible agar-agar powder (Golden Coin Agar Agar Powder, Capital Food International, Inc., La Mirada, CA, USA). The dielectric properties were measured using a network analyzer (ENA series, Keysight, Santa Rosa, CA, USA) with a high-temperature dielectric probe (85070E Kit, Agilent Technologies, Santa Clara, CA, USA). A sample of 51.3 mM saline water was used to check the measurement accuracy. The phantom’s mold was 3D printed (Appendix A) in MARTIN’s head’s shape [19]. Finally, the mold was filled with agarose solution, cured for at least 24 h, and stored in an MRI scanning room for 48 h for thermal stabilization. The 8-channel fiber optic probes (OSENSA Innovations Corp., Coquitlam, BC, Canada) were positioned at distributed locations across NeoNet including three hot spots estimated from the thermal simulation [11]. A thermal paste was used to allow the fiber optic probes to be in contact with the surface of the agar phantom and the EEG electrode to assess the RF-induced heating by the EEG net. A high-power turbo spin-echo sequence (21 slices, 0.9 × 0.9 × 5.0 mm voxels, TR/TE = 7600/86 ms, FA = 120°, Average: 20) that delivered 100% SAR for 30 min (SARhead: 2.85 W/kg, 10 gSARtorso local: 9.99 W/kg) was set to produce the maximum allowed RF safety limit in the clinical scan (IEC 60601-2-33:2010).

### 3.3. CT Quality Recordings

The CT compatibility test was conducted on a head phantom acquired in the Department of Radiology at MGH using a Siemens SOMATOM Force (Siemens Healthineers, Forchheim, Germany). The images were acquired using the Pediatric Routine Head protocol (100 kVp, eff. 66 mA, 1 s rotation time, 0.8 pitch, 35 cm DFOV, 50 cm SFOV, Hr59h kernel) performed on ’NeoNet’ while the net was positioned on a child head-sized agar phantom described above. The images of a patient with conventional EEG electrodes were found in the MGH’s PACS system.

### 3.4. MRI Quality Recordings

We imaged the three subjects in a 3 Tesla MRI (Siemens Healthineers, Germany). Since there are no commercially available pediatric nets with a label that permits scanning using the proposed clinical sequences, we used only two conditions in counterbalanced order: (1) No-Net and (2) NeoNet. Two board-certified neuroradiologists (ML, RG, each with greater than 20 years of experience reading clinical CT and MRI scans) evaluated the overall image quality of the paired NeoNet and NoNet images (three T1-weighted images, one T2-weighted image, and one diffusion tensor image), blinded to the presence or absence of the NeoNet device. A five-point Likert scale was used to score the image artifact, overall image quality, and clinical usability as follows: 1, extremely poor image quality (major artifacts exist and the images are not clinically useful); 2, poor image quality (major artifacts exist and clinical use is not advised); 3, average image quality (average moderate artifact detected, but minor consequences to clinical use); 4, good image quality (containing minor artifacts which do not adversely affect the clinical use); 5, excellent image quality (no artifacts) [32]. Images were pre-processed to remove the EEG electrodes visible outside the skull for blinding purposes. We used Infant Freesurfer [33] to perform skull stripping and create a brain mask on the T1- and T2-weighted MR imaging and diffusion tensor imaging (DTI) scans for both the NoNet and NeoNet recordings of all subjects. Subsequently, we removed all the structures that were located outside this brain mask including the skull, skin, and electrodes. The new images were randomly shown to the neuroradiologists (ML,RG) who were unable to see the presence/absence of electrodes on the skull by the masking process and were asked to score the quality of each scan based on the Likert scale, as described above. Finally, the scan ratings were compared by using a two-tailed paired *t*-test using SPSS software (IBM, Armonk, NY, USA). Results are reported as (Median, IQR), and statistical significance was defined as *p*-value < 0.05.

### 3.5. Numerical Simulations

Sim4Life (ZMT MedTech, Zurich, Switzerland) was used to solve Maxwell’s equation at 128 MHz using the FDTD method [4]. The shielded birdcage body transmit coil was designed following the realistic dimensions of a 3 T MRI [19] with a diameter of 610 mm, a spoke length of 670 mm, a shield diameter of 660 mm, and a length of 1220 mm. The coil was driven in the circularly polarized (CP) mode. The head of the 29-month-old male whole-body voxel model was positioned at the center of the body coil to assess the complex B_1_ field distribution [19]. Masks were applied on the surrounding air to display the B_1_ transmit field difference in the child model, which would otherwise not be visible because of scaling issues.

The 128-channel EEG traces were modeled following the scalp from the temporal lobe toward the parietal lobe, exiting through the top of the head (Figure 3). The 1 mm × 1 mm trace width/thickness and dielectric properties (ε_r_ = 4.2, σ = 41.7 S/m) were chosen to represent 12.23 kΩ as the nominal trace thickness was too thin to be represented in the model (i.e., 30 nm). Each trace was designed to have a minimum distance of 1 mm from each other as well as from the skin to avoid contact between the traces and the skin (Figure 3b). A locally dense grid was applied along the traces with a resolution of 0.7 mm × 0.7 mm × 1.0 mm. A high-performance graphics processing unit (GPU) (NVIDIA V10, Nvidia, Santa Clara, CA) was used to reduce the computation time. The B_1_ transmit fields were computed in four different scenarios: (i) a pediatric model without an EEG net (NoNet); (ii) with a thin film-based EEG net (NeoNet); (iii) with copper traces with an ideal current limiting resistor (R = 10 kΩ) between sponges and copper traces (CuNet); (iv) with copper traces without current limiting resistors. The B_1_ transmit field and current density was normalized to a field that produced 2 µT at the center of the coil in the case of NoNet. Then, the same input currents were applied to all other cases. The B_1_ transmit field and current density map were compared to estimate the effect of the different types of EEG traces. The difference of |B_1_^+^| between NoNet and (i) NeoNet, (ii) CuNet (copper traces with an ideal current limiting resistor), and (iii) CuNet (copper traces without current limiting resistors) were calculated as
(13)$$∆\left|B_{1}{}^{+}\right|=mean\left(\frac{{\left|B_{1}{}^{+}\right|}_{NeoNet\;or\;CuNet}-{\left|B_{1}{}^{+}\right|}_{NoNet}}{mean\left({\left|B_{1}{}^{+}\right|}_{NoNet}\right)}\times 100\right){\;}_{head}$$
where mean(|B_1_^+^|_NoNet_) is the mean value of |B_1_^+^| in the case of NoNet over the volume of the head. The uncertainty analysis of the numerical simulation was conducted as in [34]. The sensitivity factor of each parameter was calculated by running two simulations that differed only by a single parameter value of the dielectric properties of the muscle, skin, and subcutaneous adipose tissue (SAT) or fat compartments close to the EEG electrode. The second value (Value2) was set with a 10% change in the dielectric.

### 3.6. EEG Recording

The EEG recording was performed using a commercial High-Density (128 channel) EEG system (The Magstim Company Limited, Whitland, United Kingdom) and the NeoNet EEG systems on a pediatric patient with epilepsy. As per standard clinical practice, 20–40 min of EEG recordings with the commercial net were completed and followed by 10 min of EEG recording with the NeoNet EEG. One ECG electrode incorporated into the EEG cap performed the ECG recordings to prevent the ballistocardiac artifact of the EEG. A pediatric clinical neurophysiologist (J.P.) performed a qualitative comparative review of both EEG recordings to assess the quality of the background pattern and identify physiological and pathological (epileptiform) graphoelements.

## 4. Results

### 4.1. NeoNet Fabrication

The thin film traces were fabricated, and the resistance resulted in a high resistance of approximately 12.23 kΩ (Figure 4) at room temperature (i.e., 20 °C), with a minimum standard deviation (SD) of 0.49 kΩ, with a target resistance between 10 kΩ and 15 kΩ (Figure 4). The target size of the trace width, *w* = 100 µm, was based on the coupon testing (Appendix A) and the failure rates (Table 2), which increased approximately inversely proportional to *w*.

### 4.2. MRI Safety

MRI safety testing confirmed these findings with a 30-min scan, and the maximum temperature rise was found to be 0.84 °C, which is the condition that allows for 1 h of scanning without cooling time in normal operating mode as per the FDA guidelines [35] (Figure 5).

### 4.3. Image Quality

A qualitative comparison of the MR images showed a similar image quality with NoNet (Figure 6b,e) and NeoNet (Figure 6c,f), as predicted by our previous numerical simulations [36] (see Table 3). Figure 6a,d shows the T1 and T2 images with significant signal drop and distortions near the EEG metal electrodes/leads (metal artifacts). However, images of NoNet (Figure 6b,e and Figure 7 (left)) and NeoNet (Figure 6c,f and Figure 7 (right)) did not show any metal artifacts. The MR images with NeoNet and without both enabled the pediatric neuroradiologist (P.E.G.) to clinically identify the left parietal atrophy in Subject 2 indicated by the widened subarachnoid space with larger sulci and thinner gyri of the left parietal area of the brain (Figure 7 arrows on MPRAGE Subject 2) and the less prominent white matter tracts on the left side compared to the right side (Figure 7 circles on DTI Subject 2). Additionally, in Subject 3, P.E.G. also identified the left-sided tubers characterized by loss of the grey-white matter margin and somewhat irregular, suggestive of some potential areas of subtle polymicrogyria (Figure 7 MPRAGE and T2). Furthermore, when two neuroradiologists (M.H.L. and R.G.) performed a double blinded quantitative assessment on the MR images using the Likert scale, both the NeoNet and the NoNet scored (4.8, 1) with a *p*-value of 0.34, indicating no statistical difference between the two types of MR images (Table 4). A comparison of the CT images showed that the EEG had significant streak artifacts from the high attenuation metal (Figure 8a), which impeded the visualization of the CT image. While the NeoNet electrodes were also visible, no artifact from the aluminum thin film-based leads was visible (Figure 8b).

### 4.4. Numerical Simulations

The axial, coronal, and sagittal view of the absolute B_1_ transmit field was compared in Figure 9a. Figure 9c shows the difference in the absolute B_1_ transmit fields between (i) NoNet vs. NeoNet, (ii) NoNet vs. CuNet with ideal current limiting resistances, and (iii) NoNet vs. CuNet without resistors compared in the axial, coronal, and sagittal views.

Figure 9b shows the phase of the B_1_ transmit field in the axial, coronal, and sagittal views in the case of NoNet, NeoNet, and CuNet with ideal current limiting resistors and CuNet without resistors. The difference in phase of the B_1_ transmit field between (i) NoNet vs. NeoNet, (ii) NoNet vs. CuNet with ideal current limiting resistors, and (iii) NoNet vs. CuNet without resistors was also compared in Figure 9d.

The different magnitude and phase B_1_ transmit field plots, especially the difference plots, show that NeoNet had the smallest changes compared to the NoNet case, followed by the case of CuNet with a current limiting resistor, and finally without current limiting resistors.

Figure 10 shows the B_1_ transmit field profile in NoNet, NeoNet, and CuNet with and without current limiting resistors.

The current density map is displayed along the three EEG trace scenarios with a 3D surface view in Figure 11. The FDTD simulations estimated a peak root mean squared (RMS) current density of 260 kA/m^2^ in the case of CuNet without resistors, 218 kA/m^2^ in the case of CuNet with resistors, and only 9 kA/m^2^ in the NeoNet case.

The averaged magnitude of the B_1_ field and the ∆|B_1_^+^| in the head are shown in Table 3. The NeoNet had the smallest ∆|B_1_^+^| compared to the NoNet case, followed by the case of CuNet with a current limiting resistor. In contrast, the case of CuNet without current limiting resistors showed the largest ∆|B_1_^+^| compared to the case of NoNet, as expected.

The uncertainty analysis results are listed in Table 5 in terms of the estimated mean of the magnitude B_1_ transmit field. The uncertainty analysis showed a total uncertainty of 0.97% in the head’s mean magnitude B_1_ transmit field. The highest uncertainty was in the permittivity of SAT, indicating that the simulations were maximally sensitive to the fat layer in the child model.

### 4.5. EEG Recording

We recorded the EEG data on a 5-year-old male pediatric patient (Subject 4) with refractory focal epilepsy, developmental delay, and behavioral concerns. Complex partial and tonic-clonic seizures characterized his epilepsy. We recorded 10 min of resting-state brain activity with a commercial HD-EEG and the NeoNet and compared the two recordings qualitatively. Critical clinical information was detected with the NeoNet and commercial EGI HD-EEG net. The same subject of the left frontotemporal interictal epileptiform discharges was identified in both the commercial HD-EEG (Figure 12A,B) and NeoNet recordings (Figure 12C,D). Furthermore, physiological features such as muscle artifacts (Figure 13A) and sleep spindles (Figure 13B) were present, while eye movement artifacts (Figure 13C) were identified in the frontal leads. Brief, morphologically defined events characteristic of patients with epilepsy such as interictal epileptiform discharges were also identified (Figure 13D).

## 5. Discussion

### 5.1. NeoNet Fabrication

We found that the proposed thin film fabrication was superior to a previous prototype, which was manufactured using polymer thick film (PTF) with a similar technology [4]. The trace resistance was 12.23 kΩ for the thin film fabrication of 128 traces with a SD of 0.49 kΩ whereas the resistance was found to be, on average 20.49 kΩ, with a larger SD of 0.85 kΩ for 256 traces made with PTF technology. This disparity between the target and measured resistance was because the trace resistance was achieved by mixing pure silver and carbon inks. The final resistance variability depends on many factors including the mixing ratio and variables hard to control such as environmental temperature and humidity, oxidation of the mixture, and wettability of the materials, and it can only be found on the manufacturing day by test and retest. Finally, the SD for a larger number of 256 vs. 128 traces should be lower. Instead, it resulted in a larger SD for the thick film technology.

### 5.2. MRI Safety

The use of conductive EEG leads in the presence of a radio-frequency field generates induced currents on the leads [37,38] and potential increases in RF power absorbed in the human head, specified regarding the specific absorption rate (SAR) [39]. For relatively high-power RF sources such as MRI RF coils, such interactions pose serious thermal-related safety risks of tissue heating and burns [40,41,42,43,44,45]. Finite difference time domain [46] numerical estimations using a high-resolution model of head tissues [47] with EEG leads during MRI scanning simulation [37,48,49] showed the presence of local SAR peaks near the EEG electrodes, with values depending on a wide range of variables, namely, the lead orientation shielding effect of EEG leads [50], RF frequency [49,50], number of EEG electrodes/leads [49], and resistivity of the EEG leads [10,39]. These simulations, followed by temperature measurements during MR scanning with an anthropomorphic phantom, are essential for establishing system safety. Numerical simulations and temperature measurements conducted in such studies indicated that the use of metallic leads during MRI should be avoided [49] and that resistors (currently, 5–15 kΩ is the industry standard) placed on the leads do not offer reliable protection against the antenna effect of the leads exposed to the MRI RF-field [51,52]. In contrast, the increased lead resistivity afforded by PTF/thin film technology allows for safe and high-quality recordings up to 7 T [53]. With regard to EEG-fMRI safety in infants/toddlers, we could only find a single study [54] that found that on a phantom, an MRI sequence (T2 with Max Turbo factor 25; SAR 89%) caused a large temperature increase at one electrode (Fpz; +4.1 C). Additionally, this study concluded “Based on our findings, we strongly recommend against using the structural images obtained during simultaneous EEG-MRI recordings for further anatomical data analysis”. Our previous studies showed the MRI RF safety of the NeoNet using numerical simulation [19]. This study showed that the high-power MPRAGE heating was well below 2 °C. Our recent simulation study on EEG safety with an infant/toddler [55] also showed a similar finding based on the dielectric properties of the resistive traces in infants/toddlers for safe EEG-MRI [56], where thin film resistive traces reduced the RF heating at 3 T MRI within the safety guidelines by the FDA [57]. Thus, resistive traces may improve the safety of infant/toddler EEG-MRI, as shown by our SAR, and thermal safety [10,58], regardless of the sequence used. In this study, our target trace resistance of 10–15 kΩ was based on our recent study [55] in which we studied the safety using resistors of 10 kΩ, and the 5–15 kΩ industry standard. A lower resistance below 10 kΩ may affect the amount of RF-induced currents in the NeoNet traces inside a 3 T MRI, which could lead to both heating and greater B_1_ artifacts. For very total high resistances of the trace resistance and the contact EEG resistance on the subject, the EEG quality may be compromised and may damage the amplifier (the maximum total resistance allowed may vary depending on the manufacturer).

### 5.3. Numerical Simulations

Since all of the materials used in the fabrication of the NeoNet did not interfere with the B_0_ field (i.e., no susceptibility, see [59]) nor produced a chemical shift artifact, we studied the B_1_ transmit field distortion on a 29-month-old male model wearing 128-channel HD-EEG nets in a 3 Tesla MRI. The numerical simulations predicted the observed artifact-less MR images with the NeoNet, namely, the FDTD algorithm predicted a significant differential increase (with the NoNet) in the magnitude of the B_1_ transmit field of 65.2% for the CuNet with resistors and 137.5% for the CuNet without resistors compared to the NeoNet. The electromagnetic (EM) simulations were conducted on a child model to estimate the B_1_ transmit field distortion in the case of resistive EEG traces (NeoNet), CuNet with ideal current limiting resistors, and CuNet without current limiting resistors. Additionally, current density maps on three different EEG trace conductivities were compared to estimate the amount of current induced on each EEG trace due to the *antenna effect* [10,59]. The CuNet without a resistor induced a 29-fold peak RMS current density increase over the NeoNet, whereas the CuNet with ideal current limiting resistors induced a 24-fold peak RMS current density increase over the NeoNet. Therefore, the current-limiting resistor does not effectively reduce the induced current in the lead. The resulting effect is that B_1_ field distortions that these currents generate (Ampere’s law) are not affected in practice by “current limiting” resistors. In the case of CuNet without current limiting resistors, the overall B_1_ transmit field on the pediatric head was reduced due to the *shielding effect* [10,59]. Nevertheless, the extremely low B_1_ transmit field interaction of the NeoNet indicated an MRI signal almost identical to the NoNet case, which stemmed from the reduced current density induced in the resistive traces compared to the copper traces.

### 5.4. MRI, CT, and EEG Recordings

Two board-certified neuroradiologists (R.G. and M.H.L., with more than 20 years of clinical and neuroimaging research experience, respectively) independently scored the quality of MR images. Comparing the images of different sequences (T1, T2, and DTI) with NoNet and with NeoNet showed no difference in the Likert score for Subjects 1–3 (Table 4). Subject 1 was selected as a healthy adult with a small head circumference who reported that the NeoNet was comfortable enough for studies in children.

Metallic electrodes and leads may partially or entirely block X-rays. The nonlinearity caused by this phenomenon leads to a streak of low and high attenuation in the image, obscuring the underlying anatomy. Such image artifacts may obscure lesions such as small metastases. Aside from lesion identification, metallic implants may interfere with radiation delivery by blocking it from reaching a deeper target. No quantitative Likert score was performed for CT scans since neuroradiologists are not used to evaluate phantom images. However, the absence of CT artifacts with the NeoNet was evident in all slices due to the low-density materials (e.g., aluminum) used in the NeoNet that avoided streak artifacts.

Regarding EEG metal artifacts, Figure 6a (blue arrow) shows an increase in signal intensity as a B_1_ artifact predicted by the simulations (e.g., Figure 9a bottom arrow). In contrast, Figure 6d (blue arrow) shows a decrease in signal intensity as a B_1_ artifact predicted by the simulations (e.g., Figure 9a top arrow). In part, these artifacts could also be susceptibility artifacts due to the different magnetic properties of the metals with respect to the human tissue [59].

Comparative EEG data recorded with a commercial HD-EEG net and the NeoNet showed a similar quality in both recordings. Physiological EEG features were clinically identified in both recordings by a neurologist (J.M.P.), and the same population of interictal epileptiform discharges was identified in the recording of the same subject, in support of the clinical utility of the NeoNet EEG signal compared to commercially available net(s).

### 5.5. Limitations

#### 5.5.1. Manufacturing

In order to reach the target resistance of R_t_, we had to design five loops per trace since the resistivity was lower than what we ideally needed. We could have used thinner, higher-resistivity aluminum metalized films or other metals such as titanium (i.e., higher resistivity, non-magnetic, biocompatible, etc.). In all cases, we were not able to contract or manufacture our own metalized polyimide thin film because we were unable to find a roll-to-roll e-beam fabrication facility appropriate for this application.

#### 5.5.2. Numerical Simulations

The high-frequency lumped circuit model of the resistors including series inductors and parallel capacitors was not included in these numerical simulations. The EEG traces were treated as open-ended circuits, which might have differed when the EEG was connected to the amplifiers during the MRI experiment. The effect of the EEG traces with different trajectories was not numerically simulated nor validated in this study.

#### 5.5.3. EEG Recording

In the prototype model, several electrodes produced flat channels in the bipolar montage. This was due to the electrical instability of the carbon conductive glue (Appendix A), which, in some electrodes, was subsequently fixed by coating the aluminum layer with 10 nm of Cr and 300 nm of Au (Figure 2) using e-beam deposition at the Center for Nanoscale Systems to improve the chemical resistance of the electrodes against exposure to the hydrating saline (e.g., KCl) and disinfectant solutions. The Ag/AgCl disks were glued to the aluminum/gold electrodes using a silver conductive epoxy adhesive (MG Chemicals, Burlington, ON, Canada) instead of the electrically unstable carbon glue.

## 6. Conclusions

Metallic artifacts and distortions can substantially degrade the image quality of both MRI and CT when EEG electrodes are present. Thus, we showed that the novel thin film trace EEG net (“NeoNet”) resulted in improved MRI and CT image quality without compromising the EEG signal quality. The 50 cm long thin film trace was constructed from a 30 nm aluminum film, resulting in low density for reducing CT artifacts and a 12 kΩ high-resistive low for reducing MRI artifacts. The NeoNet demonstrated safety in 3 Tesla (T) MRI for children with an increase of just 0.84 °C after 30 min of high-power scanning, which is the acceptance criteria for the temperature for 1 h of normal operating mode scanning as per the FDA. We investigated the effects of EEG nets on B1 transmit field distortion in 3 T MRI with electromagnetic simulations, which predicted a 65% B1 transmit field distortion higher for commercially available copper-based EEG nets over the NeoNet.

No significant difference in the degree of MRI artifact or image distortion was found while analyzing the Likert scale responses from two board-certified neuroradiologists blinded to the presence or absence of NeoNet. Finally, NeoNet did not impact the quality of the EEG recording and allowed 128 dense EEG spatial sampling channels for clinical epilepsy diagnostics.

The proposed NeoNet device, therefore, has the potential to allow concurrent EEG acquisition and MRI or CT scanning without significant image artifacts, facilitating clinical care and EEG/fMRI pediatric research.

## Figures and Tables

**Figure 1 sensors-23-03633-f001:**
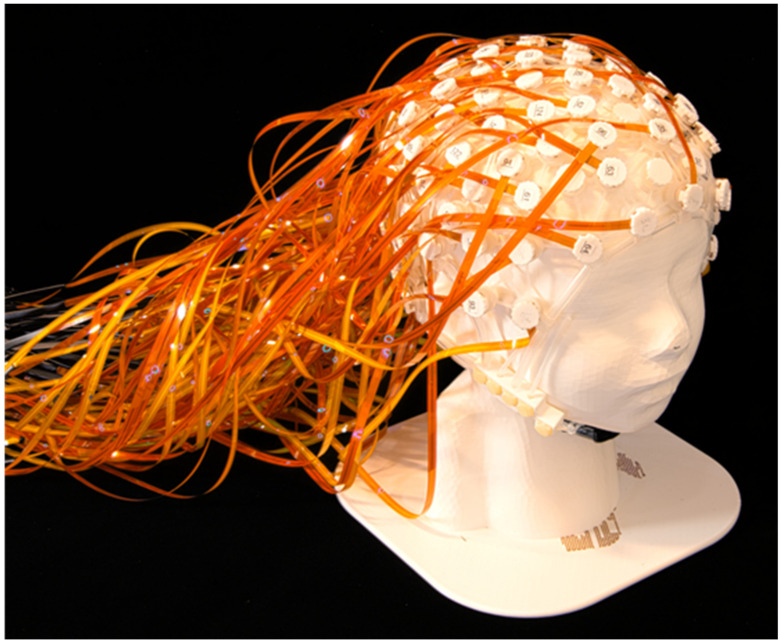
The NeoNet. A 128-channel high-density EEG net based on aluminum thin film traces.

**Figure 2 sensors-23-03633-f002:**
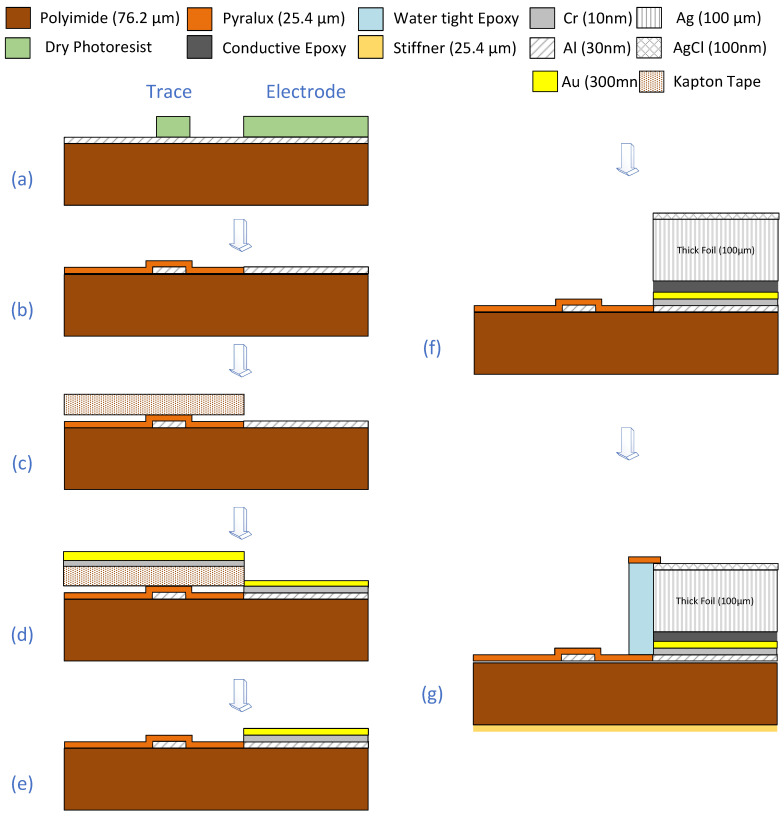
The fabrication process flowchart for the NeoNet traces and electrodes. (**a**) Photolithography of the Al metalized polyimide. (**b**) Trace insulation. (**c**) Masking. (**d**) E-beam deposition. (**e**) Wet etching. (**f**) Electrode assembly. (**g**) Electrode waterproofing.

**Figure 3 sensors-23-03633-f003:**
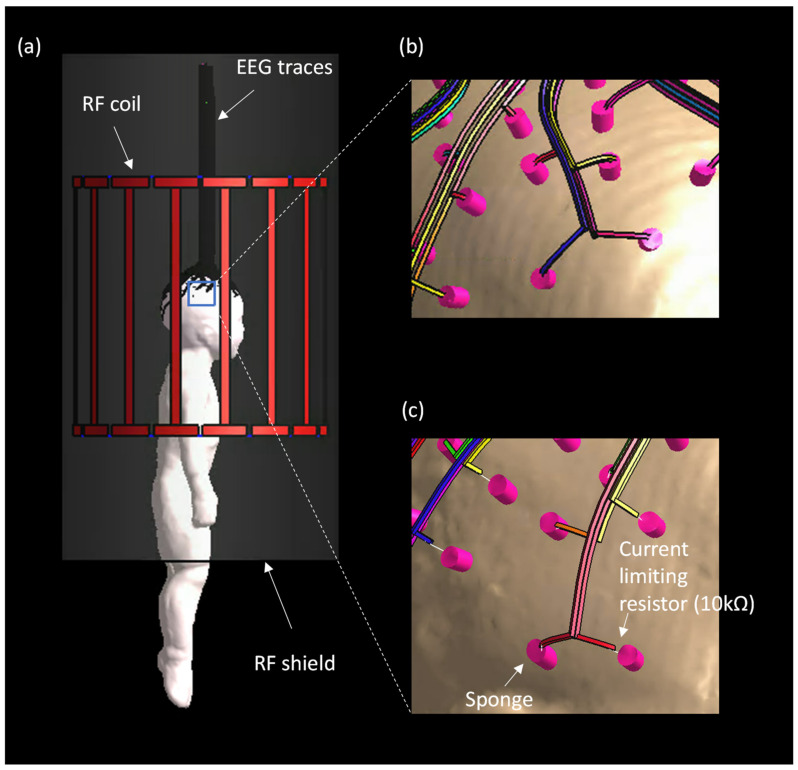
Geometry of the numerical simulations. (**a**) Geometry of the 29-month-old male model positioned at the center of a shielded body transmit coil wearing the NeoNet, (**b**) geometry of the EEG traces without current limiting resistors, (**c**) EEG traces with current limiting resistors.

**Figure 4 sensors-23-03633-f004:**
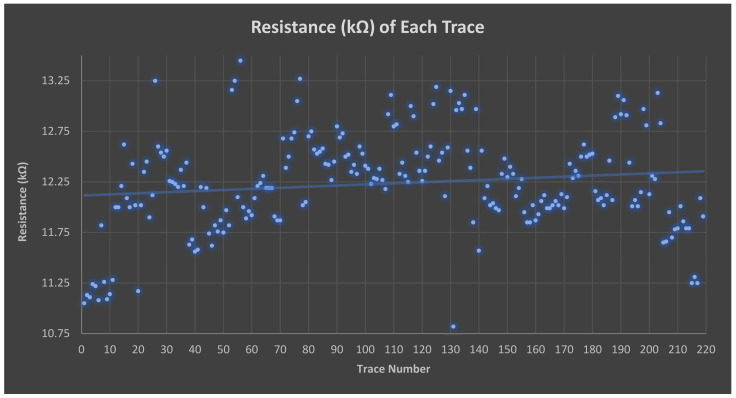
Trace-by-trace resistance measurement.

**Figure 5 sensors-23-03633-f005:**
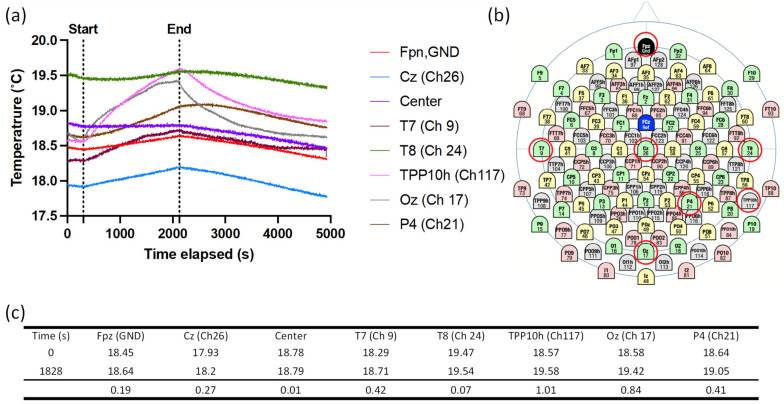
3 T MRI RF safety test results of the NeoNet with a child head phantom. (**a**) Temperature elevation with a high-power turbo spin-echo sequence with a maximum allowed input power in a clinical scan for 30 min using a birdcage body transmit coil. (**b**) Distribution of 128-channel EEG traces and the position of EEG electrodes where the temperature was monitored. (**c**) Table of relative temperature changes in the location of the electrodes monitored (red circle) over 30 min of scanning.

**Figure 6 sensors-23-03633-f006:**
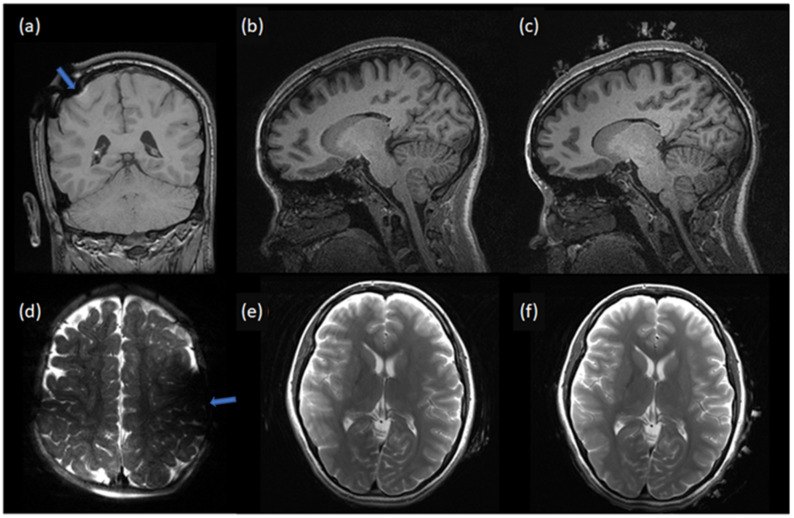
MRI data with EEG electrodes vs. NoNet vs. NeoNet. (**a**–**c**) Comparison of the T1 and (**d**,**e**) T2 sequences of adult volunteers. Images (**a**,**d**) taken with EEG electrodes show the presence of a B_1_ artifact (blue arrow), whereas no artifacts were visible with both NoNet (**b**,**e**) and NeoNet (**c**,**f**).

**Figure 7 sensors-23-03633-f007:**
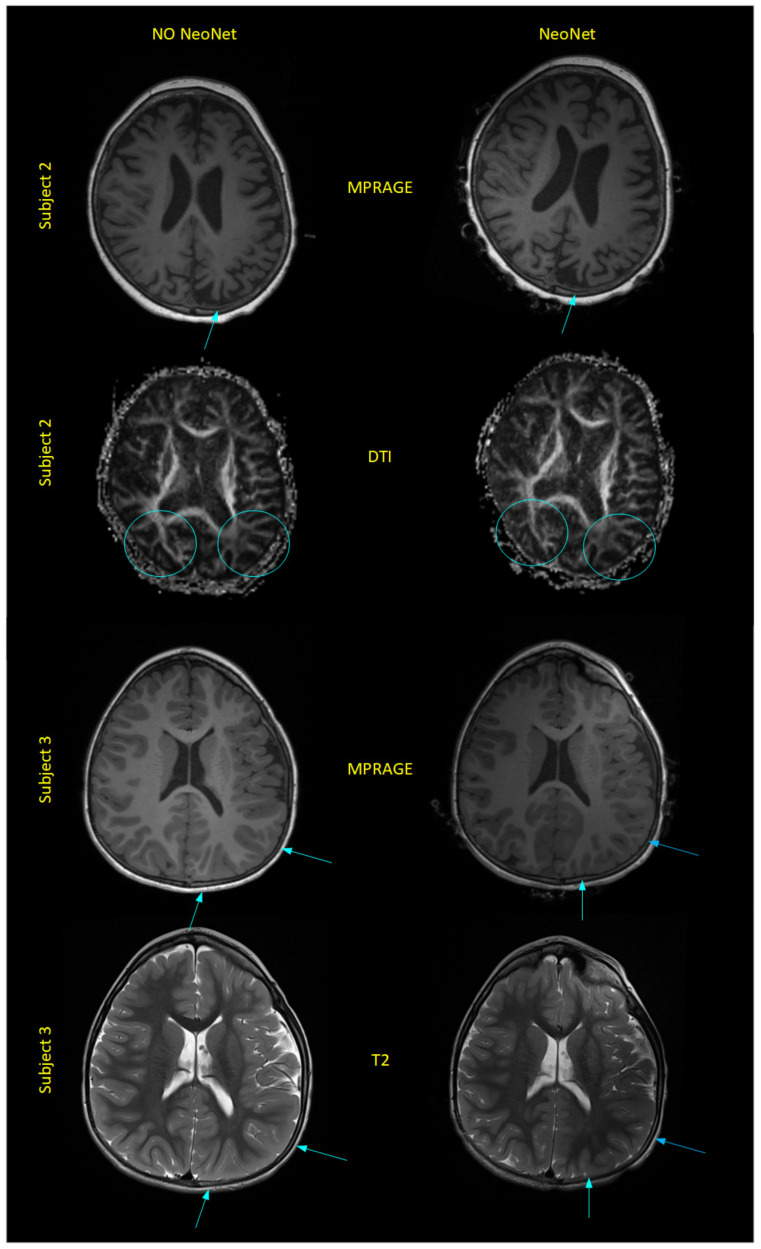
MRI data with NoNet (**left**) vs. NeoNet (**right**). Both the NoNet (**left**) and NeoNet (**right**) enabled the pediatric neuroradiologist P.E.G. to clinically identify the lesions. Subject 2: Comparison of T1 magnetization prepared rapid gradient echo (MPRAGE) and diffusion tensor imaging (DTI) sequences of a five year old female with left-side parietal atrophy (indicated by arrows on the MPRAGE and circles on the DTI that point to the differences in the white matter tracts between the left and the right hemisphere). Subject 3: Comparison of MPRAGE and T2 sequences of a five year old male with left-side tuberous malformations (arrow) showed a similar image quality of NoNet (**left**) and with NeoNet (**right**).

**Figure 8 sensors-23-03633-f008:**
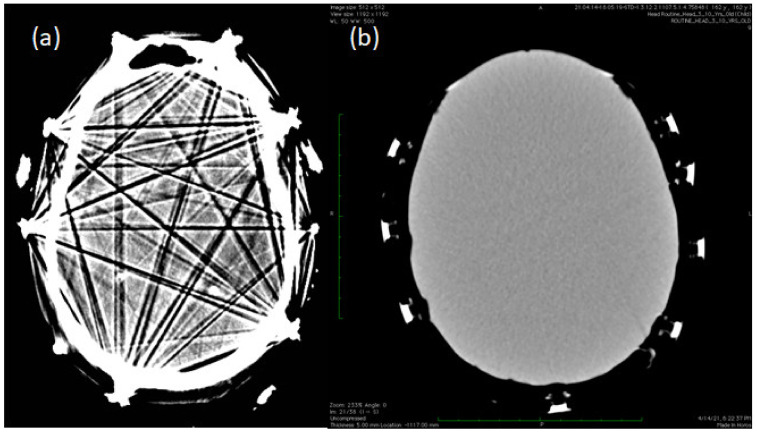
Axial CT images with conventional EEG electrodes (left) vs. Neo-Net EEG (right). (**a**) Axial view of a patient with artifacts from EEG electrodes. (**b**) Images with the NeoNet without noticeable artifacts from electrodes inside the phantom. In both cases, similar display settings were used for the window-width and center-level.

**Figure 9 sensors-23-03633-f009:**
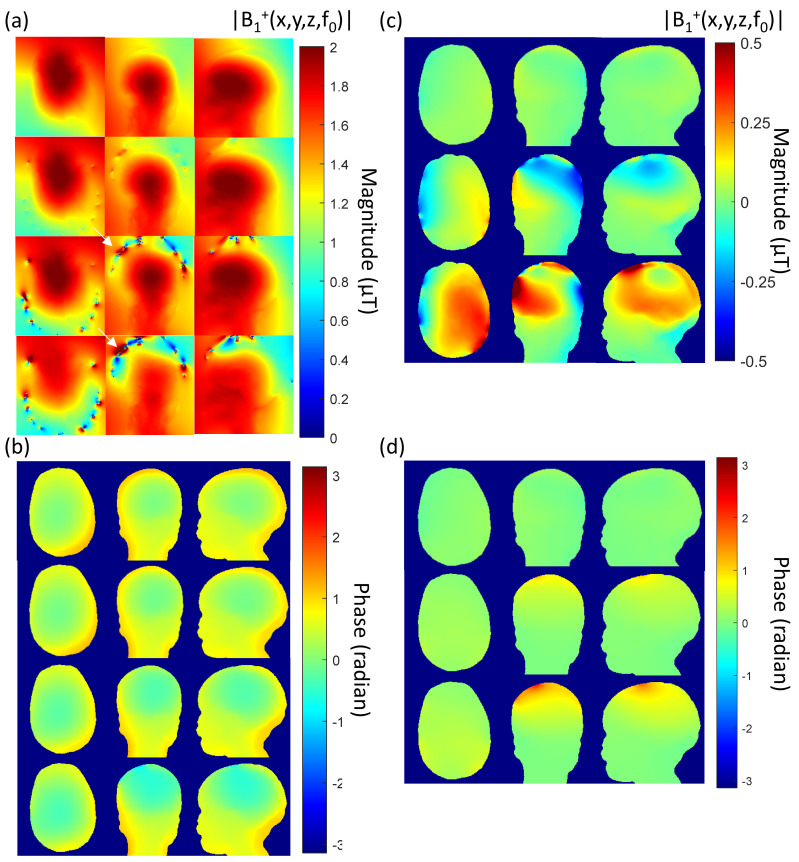
(Left column) (**a**) Simulated B_1_ transmit field distribution magnitude and (**b**) phase; axial (left), coronal (middle), and sagittal (right) view in the case of NoNet, NeoNet, and CuNet with ideal current limiting resistors, and CuNet without resistors. Black arrows indicate a decrease (top) and an increase (bottom) in the local B_1_ field magnitude. (Right column) (**c**) Difference in the B_1_ transmit field distribution magnitude and (**d**) phase; axial (left), coronal (middle), and sagittal (right) view in the case of (top) NoNet–NeoNet, (middle) NoNet–CuNet with ideal current limiting resistors, and (bottom) NoNet–CuNet without resistors.

**Figure 10 sensors-23-03633-f010:**
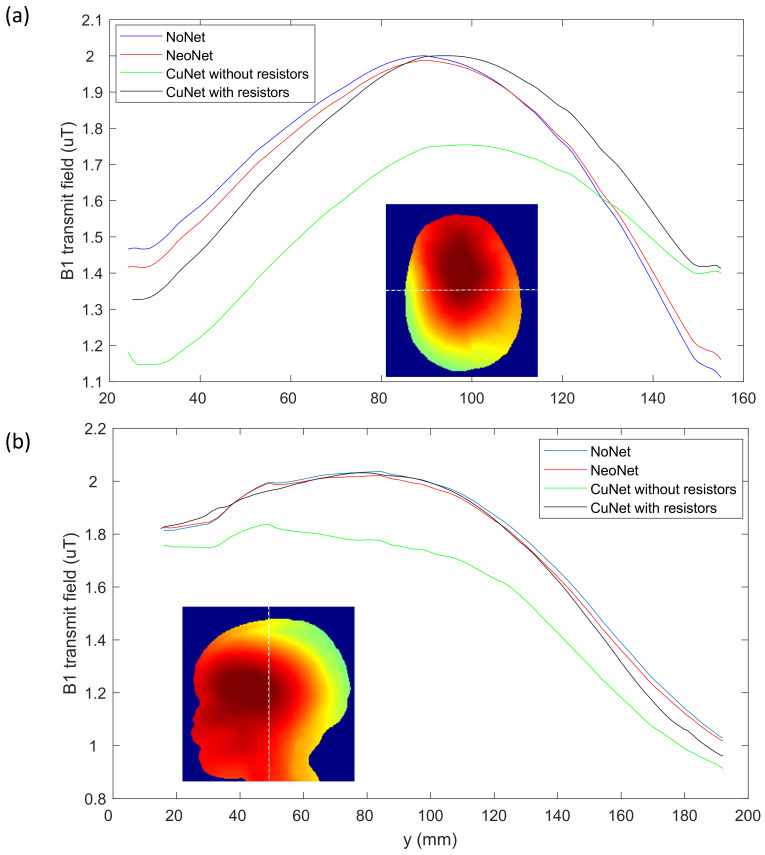
Comparison of the B_1_ transmit field among four cases with dashed lines of the position of the profile. (**a**) Axial profile and (**b**) longitudinal profile in the central slices.

**Figure 11 sensors-23-03633-f011:**
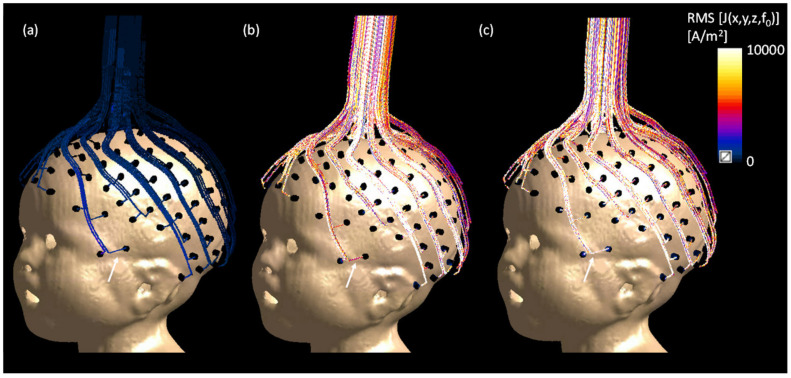
Current density map displayed on EEG traces in the 3D surface view. (**a**) NeoNet, (**b**) CuNet with ideal current limiting resistors, (**c**) CuNet without resistors.

**Figure 12 sensors-23-03633-f012:**
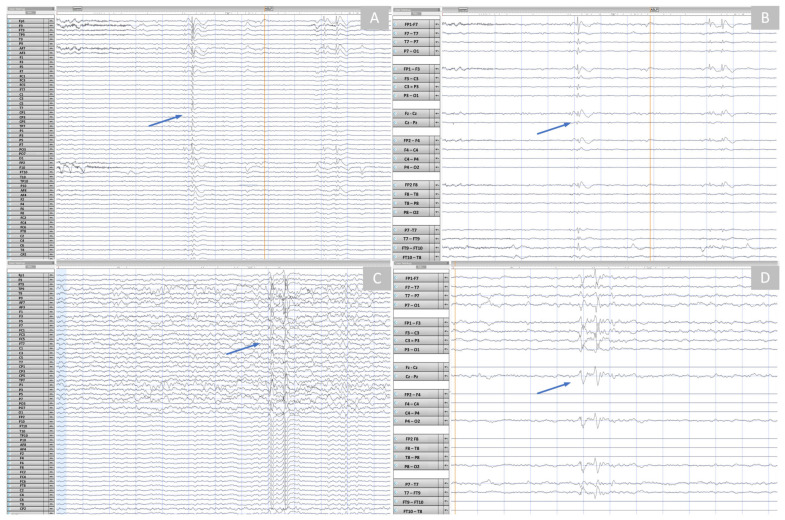
EEG Recording (outside the MRI) using the NeoNet vs. a commercial Net while the pediatric patient is asleep. (**A**) EEG screenshot from EGI (commercial) HD-EEG using 10–10 channel labels, system reference. Left frontotemporal spikes were visible. (**B**) EEG screenshot from EGI HD-EEG with spatially downsampled EEG to the 10–20 electrode positions, anterior–posterior bipolar reference. Note the complex field of the spike. (**C**) EEG screenshot from NeoNet EEG using 10–10 channel labs, system reference. The same left frontotemporal spike population was evident. (**D**) EEG screenshot from NeoNet EEG with spatially downsampled EEG to the standard 10–20 electrode positions, anterior–posterior bipolar reference. The same spike was shown (arrows). In the bipolar montage, the right-sided channels that carry the same signal (see (**C**)) were now flat.

**Figure 13 sensors-23-03633-f013:**
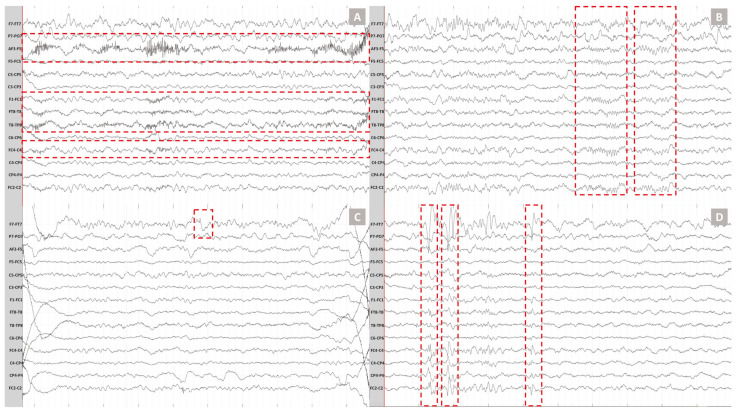
Physiological and pathological signals recorded with the NeoNet EEG system in a 5-year-old patient with epilepsy. The dotted red lines indicate (**A**) the muscle artifact in bilateral temporal extending to the frontocentral leads. (**B**) Sleep spindles at frontocentral leads with their maximum at the FC2-C2 and FC4-C4 leads. (**C**) Eye movement artifact on the left frontal channel. (**D**) Interictal epileptiform discharges with complex field originating from the frontal EEG leads.

**Table 1 sensors-23-03633-t001:** Physical properties of biocompatible and non-magnetic metals.

	Conductivity (S/m)	Resistivity (Ω·m)	Mass Density (Kg/m^3^)	α (Ω/C)
Aluminum	3.7 × 10^7^	2.65 × 10^−8^	2700	4.3 × 10^−3^
Copper	5.96 × 10^7^	1.68 × 10^−8^	8930	3.9 × 10^−3^
Gold	4.1 × 10^7^	2.44 × 10^−8^	19,302	3.7 × 10^−3^
Titanium	1.8 × 10^6^	5.56 × 10^−7^	4500	4.7 × 10^−3^

**Table 2 sensors-23-03633-t002:** Failure rates at different trace widths of the coupons in Appendix A.

Trace Width (″)	0.004	0.0035	0.003	0.0025	0.002	0.0015	0.001	0.0008
Resistance (kΩ)	11.3	13.1	15.2	18.7	23.8	31.3	24.0	N.E.
STDEV (kΩ)	±0.29	±0.27	±0.42	±0.44	±0.57	±1.12	N.E.	N.E.
Failure Rate (%)	21.0	25.0	37.5	37.5	50.0	71.0	96.0	100.0

**Table 3 sensors-23-03633-t003:** B_1_ transmit field uniformity from Sim4Life Numerical Simulations on MARTIN [19].

	NoNet	NeoNet	CuNet (With Ideal Resistors)	CuNet (Without Resistors)
Mean B_1_^+^ field (µT)	1.54	1.53	1.57	1.46
∆|B_1_^+^| (%)	-	0.06	3.91	8.25

**Table 4 sensors-23-03633-t004:** Likert scale results from two board-certified neuro-radiologists comparing MR images with the NeoNet and NoNet conditions. The NoNet and NeoNet images were masked off to exclude skin and structures outside the skin to avoid the bias of detecting the presence of the EEG electrodes. Furthermore, the images of NeoNet and NoNet were double blinded and randomly presented to the neuro-radiologists.

	Scan	NeoNet EEG	Reviewer 1	Reviewer 2
Subject 1	T1	No	4	4.5
	T1	Yes	4	4.5
Subject 2	T1	No	5	5
	DTI	No	4	5
	T1	Yes	5	5
	DTI	Yes	4	5
Subject 3	T1	No	5	5
	T2	No	4	4.5
	T1	Yes	5	5
	T2	Yes	4	4

**Table 5 sensors-23-03633-t005:** Uncertainty of the mean B_1_ transmit field estimation. The methods used were based on the work of Neufeld et al. [34] to evaluate the uncertainty of the quantities derived by simulation; two simulations were assessed for each parameter by assigning two different values (“Value1” and “Value2”). The first value (“Value1”) was the one used for the simulation, whereas the second value (“Value2”) was set to a realistic value that could occur from the measurement or design choice. The results obtained for each value (“Result1”, and “Result2”) were used to evaluate the sensitivity factor of the quantity evaluated (mean B_1_ transmit field). The standard deviation (“Std. Dev.”) was derived from literature.

Mean B_1_^+^	Uncertainty of the Mean B_1_ Transmit Field Estimation						
	Val1	Val2	Result1 (µT)	Result2 (µT)	Sensitivity Factor [%/%]	Std. Dev. [34]	Uncertainty (%)
Muscle cond. [S/m]	1.01	0.91	1.533	1.534	0.0021	0.04	0.01
Muscle perm.	74.92	67.43	1.533	1.530	0.0189	2.80	0.07
Skin cond. [S/m]	0.78	0.71	1.533	1.535	0.0124	0.04	0.06
Skin perm.	84.41	76.00	1.533	1.529	0.0253	2.80	0.08
SAT cond. [S/m]	0.10	0.09	1.533	1.534	0.0082	0.04	0.33

## Data Availability

The data that support the findings of this study are available from the corresponding author upon reasonable request.

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
