# Peer review of "Aluminum Thin Film Nanostructure Traces in Pediatric EEG Net for MRI and CT Artifact Reduction"

_sensors, 2023, doi:10.3390/s23073633_

Round 1
Reviewer 1 Report
The authors have undertaken a substantial amount of labor to fabricate a novel EEG acquisition instrument. The content and accompanying materials are very detailed, however, the format necessitates refinement, particularly in regards to the arrangement of figures. The placement or labeling of Figure 4, in particular, requires adjustment. In totality, the quality of this manuscript is suitable for publication. I would like to recommend it for publication in Sensors.
Author Response
The authors have undertaken a substantial amount of labor to fabricate a novel EEG acquisition instrument. The content and accompanying materials are very detailed, however, the format necessitates refinement, particularly in regards to the arrangement of figures. The placement or labeling of Figure 4, in particular, requires adjustment. In totality, the quality of this manuscript is suitable for publication. I would like to recommend it for publication in Sensors.
Re: We thank the Reviewer for our manuscript's careful and insightful review. We thank the Reviewer for the positive comment. Recommending acceptance and highlighting the completeness and the very detailed materials in the paper. The suggested changes in the arrangement of the figures and the relative refinements have greatly improved our paper. We have combed very carefully throughout the entire manuscript to ensure that the figure numbering was in the correct sequence and the labeling closely described the relative figure.
Reviewer 2 Report
Comments of this reviewer on the manuscript Sensors-2280871 are as follows:
1. According to https://doi.org/10.1148/rg.2018170102 and other relevant literature, the following “Artifacts caused by metallic implants appear as dark and bright streaks at computed tomography (CT), which severely degrade the image quality and decrease the diagnostic value of the examination.” is well-known. Accordingly, this manuscript should provide images showing how CT and MRI artifacts are reduced using the proposed NeoNet system having long, flexible, and narrow aluminum thin film traces. According to the Authors, this also should be the main contribution of the manuscript. From Figure 6, as well as from other content of this submission, it cannot be concluded that the application of the proposed NeoNet system resulted in any reduction of some CT and MRI artifacts.
2. From the content of this manuscript, it is obvious that the manuscript represents a novel product, that is, a novel 128-channel high-density EEG net based on thin-film traces. In a research article, any novel product can be presented, but from a scientific point of view, and not only through some descriptions, test results, and discussions. Therefore, this manuscript lacks the scientific purpose and scientific contributions.
3. Having in mind the above-given description related to artefacts, the correctness of the following sentence: “Metal artifacts may decrease lesion identification for radiation diagnosis and reduce the accuracy of radiotherapy's target delineation and dose calculation.” might be questionable. What is the meaning of the term “metal artifact”? The use of the term “artifact” must be identical at the level of this study as well as the associated research field in general.
4. Related to the trace resistance, the following can be found: “The SD of trace resistance for the thin-film fabrication of traces was 0.49kOhm. Whereas a previous prototype was manufactured using Polymer Thick Film (PTF) with a similar technology [35], the resistance was found to be, on average 20.49 kOhm, with a larger SD of 0.85 kOhm for 256 traces.”, “The thin-film traces were fabricated, and the resistance was found to be, on average 12.23 kOhm, with a minimum Standard Deviation (SD) of 0.49 kOhm, with a target resistance between 10 kOhm and 15 kOhm (Figure 4).” and so on. In this regard, it is clear that the trace resistance should be between 10 kOhm and 15 kOhm. However, it is not explained why. What would happen if this resistance is higher or lower than the target resistance values?
5. Sub-section 5.2 has the form of a literature review, not the form of any discussion of the obtained results.
6. In the following “All images were appropriate for clinical reading as evaluated by the two neuroradiologists, and image quality was not affected by the application of the NeoNet EEG during the recording.”, the two neuroradiologists found that the image quality was not affected by the application of the NeoNet system. This is far away from the standardized scientific approaches.
7. The conclusions do not have any scientific significance, and must be reorganized taking into account the scientific contributions, if any.
8. What is the relevant state-of-the-art for the topic considered? Some of the references used in this manuscript are ancient.
9. Figures 2 and 3 appear in the text after Figure 4. They should appear after their first mentions in the text.
10. The Authors must find a way to include supplementary material in the work, and only the material which has scientific significance.
11. Keywords should be listed in alphabetical order.
12. There are some typos in this manuscript. For instance: “…applications Of the…”, “…artifacts., thus traces…”, etc.
13. Some abbreviations such as “GPU” or “TSE” are used without the associated descriptions.
14. The presentation style of this manuscript must be completely modified from the aspect of the scientific contribution.
Author Response
We address all of the concerns of the reviewer as follows.
- According to https://doi.org/10.1148/rg.2018170102 and other relevant literature, the following “Artifacts caused by metallic implants appear as dark and bright streaks at computed tomography (CT), which severely degrade the image quality and decrease the diagnostic value of the examination.” is well-known. Accordingly, this manuscript should provide images showing how CT and MRI artifacts are reduced using the proposed NeoNet system having long, flexible, and narrow aluminum thin film traces. According to the Authors, this also should be the main contribution of the manuscript. From Figure 6, as well as from other content of this submission, it cannot be concluded that the application of the proposed NeoNet system resulted in any reduction of some CT and MRI artifacts.
Re: We would like to thank Reviewer for taking the necessary time and effort to review the manuscript. We sincerely appreciate all your constructive and detailed comments, which helped us in improving the quality of the manuscript. The new figures show now common EEG induced artifacts both in MRI and CT, which were missing in the previous version of the paper or were scarcely visible. The examples were taken from The Massachusetts General Hospital Raioology (PACS) archive, so thery represent images from actual clinical cases, albeit rare cases. In general EEG are almost always removed before imaging at MGH and in many tier Hospitals, because of the potential artifacts (even barely visible to the untrained eyes, as per our previous examples). Artifacts in the MRI or CT would require to retake the examination after removing the EEG leds, which would create financial losses and diagnostic delays, so there are only a limited number of MRI/CT studies available in PACS.
The following are the new Figures inserted in this paper to better visualize the artifacts induced by the EEG:
|
|
|
|
|
Figure 6: MRI data with EEG electrodes vs. no net vs. NeoNet. (a-c) Comparison of T1, and (d-e) T2 sequences of adult volunteers show the presence of a B1 artifact (blue arrow) with EEG electrodes (a,d), NoNet (b,e), and NeoNet (c,f).
|
|
|
|
Figure 8: CT phantom data of NeoNet (left) vs. Cu-Net (right).(bad clinical -> Phantom with commercial net but with contrast that makes artifact look worse-> NeoNet) These CT data were acquired using the Siemens SOMATOM Force. (a) shows the front view of the commercial EEG net (EGINet) with the child head phantom position on a CT scanner, (b) shows the NeoNet with child head phantom on the patient bed of the CT scanner, (c) shows the CT localizer results of EGINet with red arrows indicating the EEG trace shown in the image, (d) axial view of the 'WO 5 x 5' CT sequence of EGI net with blue arrows indicating the artifacts of the EEG electrodes, (e) shows the localizer of the NeoNet without signal of the thin-film based EEG traces, and (f) shows the axial view of the WO 5x5 CT sequence without noticeable artifacts from electrodes inside the phantom.
|
- From the content of this manuscript, it is obvious that the manuscript represents a novel product, that is, a novel 128-channel high-density EEG net based on thin-film traces. In a research article, any novel product can be presented, but from a scientific point of view, and not only through some descriptions, test results, and discussions. Therefore, this manuscript lacks the scientific purpose and scientific contributions.
Re: The authors respectfully disagree with the Reviewer that the manuscript represents a novel product. In all honesty and after discussion with the authors, especially Drs Warbrick and Jaschke from Brain Products, the NeoNet is not being considered as a product given the extreme high cost and complexity in the manufacturing process. Nevertheless, after 3 years of attempts we were able to manufacture it quite successfully. In this respect, this manuscript represents a substantial advance in nanoscale fabrication, which is at the forefront in the science and technology of sensors. We show that we were able to develop a fabrication process of ultra-high-aspect-ratio of 17,000:1 (30nm×50.8cm×100µm). This is quite an achievement, since normally in nanoscale fabrication, structures are built with aspect rations of 5:1 or 10:1, and are considered high aspect ratios. This process may allow to develop future sensors like the one in the manuscript, that go beyond the usual limitations in wafer size used in micro and nano fabrication and at the same time allow for flexible substrates. We have now introduced a figure that best describes in detail this manufacturing process:
|
|
|
Figure 2: The fabrication process flowchart for the NeoNet traces and electrodes. (a) Photolithography of the Al metalized polyimide. (b) Trace insulation. (c) Masking. (d) Ebeam deposition. (e) Wet etching. (f) Electrode assembly. (g) Electrode waterproofing. |
This process was developed after careful measuremnent of failure tests now available in a Table not present in the main manuscript before:
|
Trace width (") |
0.004 |
0.0035 |
0.003 |
0.0025 |
0.002 |
0.0015 |
0.001 |
0.0008 |
|
Resistance (kΩ) |
11.3 |
13.1 |
15.2 |
18.7 |
23.8 |
31.3 |
24.0 |
N.E. |
|
STDEV (kΩ) |
± 0.29 |
± 0.27 |
± 0.42 |
± 0.44 |
± 0.57 |
± 1.12 |
N.E. |
N.E. |
|
Failure Rate (%) |
21.0 |
25.0 |
37.5 |
37.5 |
50.0 |
71.0 |
96.0 |
100.0 |
|
Table 2: Failure rates at different trace widths. |
||||||||
Furthermore, another table not present before in the manuscript, now reports Likert scale results of radiological image quality:
|
|
Scan |
NeoNet EEG |
Reviewer 1 |
Reviewer 2 |
|
Subject 1 |
T1 |
No |
4 |
4.5 |
|
T1 |
Yes |
4 |
4.5 |
|
|
Subject 2 |
T1 |
No |
5 |
5 |
|
DTI |
No |
4 |
5 |
|
|
T1 |
Yes |
5 |
5 |
|
|
DTI |
Yes |
4 |
5 |
|
|
Subject 3 |
T1 |
No |
5 |
5 |
|
T2 |
No |
4 |
4.5 |
|
|
T1 |
Yes |
5 |
5 |
|
|
T2 |
Yes |
4 |
4 |
Table 4: Likert Scale results from two board-certified neuro-radiologists
Importantly, the images taken with the NeoNet were used for clinical diagnostic, as shown now in the new Figure 7:
|
|
|
Figure 7: MRI data with NoNet (left) vs. NeoNet (right). Subject 2: Comparison of T1 Magnetization Prepared Rapid Gradient Echo (MPRAGE) and diffusion tensor imaging (DTI) sequences of a five years old female with left-side parietal atrophy (indicated by arrows on the MPRAGE and Circles on the DTI that point to the differences of the white matter tracts between the left and the right hemisphere), shows the similar image quality of NoNet (left) and with NeoNet (right) Subject 3: Comparison of MPRAGE, and T2 sequences of a five years old male with left side tuberous malformations (Arrow), shows the similar image quality of NoNet (left) and with NeoNet (right) |
These new MRI clinical data was also enriched by EEG clinical data, now included in the main manuscript:
|
|
|
Figure 13: Physiologic and pathologic signals recorded with the NeoNet EEG system. (A) Muscle artifact in bilateral temporal extending to the frontocentral leads. (B) Sleep spindles at frontocentral leads with their maximum at FC2-C2 and FC4-C4 leads. (C) Eye movement artifact (D) Interictal epileptiform discharges.
|
Finally, new data on the numerical simulations show how the profile of the B1 field varies with the various types of EEG nets:
|
|
|
Figure 10: Comparison of the B1 transmit field among four cases with dashed lines of position of profile. (a) axial profile and (b) longitudinal profile in the central slices. |
All the numerical simulations are supported by the so called uncertainty Table, which is produced a large number of simulations to study the robustness and sensitivity of the FDTD simulations. Now available also in the main manuscript:
|
Mean B1+ |
Uncertainty of the mean B1 transmit field estimation. |
||||||
|
Val1 |
Val2 |
Result1 (µT) |
Result 2 (µT) |
Sensitivity Factor [%/%] |
Std. Dev. [2] |
Uncertainty (%) |
|
|
Muscle cond. [S/m] |
1.01 |
0.91 |
1.533 |
1.534 |
0.0021 |
0.04 |
0.01 |
|
Muscle perm. |
74.92 |
67.43 |
1.533 |
1.530 |
0.0189 |
2.80 |
0.07 |
|
Skin cond. [S/m] |
0.78 |
0.71 |
1.533 |
1.535 |
0.0124 |
0.04 |
0.06 |
|
Skin perm. |
84.41 |
76.00 |
1.533 |
1.529 |
0.0253 |
2.80 |
0.08 |
|
SAT cond. [S/m] |
0.10 |
0.09 |
1.533 |
1.534 |
0.0082 |
0.04 |
0.33 |
|
SAT perm. |
15.09 |
13.58 |
1.533 |
1.530 |
0.0228 |
2.80 |
0.42 |
|
Table 5. Uncertainty of the mean B1 transmit field estimation: The methods used were based on the work of Neufeld et al. [2] to evaluate the uncertainty of the quantities derived by simulation; two simulations were assessed for each parameter by assigning two different values ("Value 1" and "Value 2"). The first value ("Value1") was the one used for the simulation shown in Table I, whereas the second value ("Value2") was set to a realistic value that could occur from measurement or design choice. The results obtained for each value ("Result1", and "Result 2") were used to evaluate the sensitivity factor of the quantity evaluated (mean B1transmit field). The standard deviation ("Std. Dev.”) was derived from the literature. |
|||||||
In regards to the scientific premise, we have added the principal rationale for developing such technology, basically Radiology departments all around the country, including MGH and BCH prefer to take out EEG nets before imaging because of artifacts.
The introductions now specifies:
“However, typically EEG nets must be removed prior to MRI and CT imaging, as they can pose a safety hazard in the MRI environment and can severely impact image quality The removal and re-application process of the EEG net is time-consuming and labor-intensive, and can result in long periods where the EEG cannot be monitored, potentially impacting patient care and limiting health care teams’ ability to make well-informed decisions.
”
- Having in mind the above-given description related to artefacts, the correctness of the following sentence: “Metal artifacts may decrease lesion identification for radiation diagnosis and reduce the accuracy of radiotherapy's target delineation and dose calculation.” might be questionable. What is the meaning of the term “metal artifact”? The use of the term “artifact” must be identical at the level of this study as well as the associated research field in general.
Re: The Reviewer brought up a very interesting point. In the manuscript, we have linked the observed artifacts with the prediction from the simulations, and both (images and numerical predictions) are pointed by arrows in the two figures:
|
|
|
|
|
Figure 6: MRI data with EEG electrodes vs. no net vs. NeoNet. (a-c) Comparison of T1, and (d-e) T2 sequences of adult volunteers show the presence of a B1 artifact (blue arrow) with EEG electrodes (a,d), NoNet (b,e), and NeoNet (c,f).
|
|
|
|
Figure 9: (Left Column) (a) Simulated B1 transmit field distribution magnitude and (b) phase; Axial (left), Coronal (middle), and Sagittal (right) view in case of NoNet, NeoNet, CuNet with ideal current limiting resistors, and CuNet without resistors. Black arrows indicate a decrease (top) and an increase (bottom) in local B1 field magnitude. (Right column) (c) Difference in B1 transmit field distribution magnitude and (d) phase; Axial (left), Coronal (middle), and sagittal (right) view in case of (top) NoNet - NeoNet, (middle) NoNet – CuNet with ideal current limiting resistors, and (bottom) NoNet – CuNet without resistors.
|
The discussion now includes the following text:
“Metallic electrodes and leads may partially or entirely block X-rays. The non-linearity caused by this phenomenon leads to a streak of low and high attenuation in the image, obscuring the underlying anatomy. Such image artifacts may obscure lesions such as small metastases. Besides lesion identification, metallic implants may interfere with radiation delivery by blocking it from reaching a deeper target. No quantitative Likert score was performed for CT scans since neuroradiologists are not used to evaluate phantom images. However, the absence of CT artifacts with the NeoNet was evident in all slices due to the low-density materials (e.g., aluminum) used in the NeoNet that avoided streak artifacts.
Regarding EEG metal artifacts, Figure 6a (blue arrow) shows an increase in signal intensity as a B1 artifact predicted by the simulations (e.g., Figure 9a bottom arrow). In contrast, Figure 6d (blue arrow) ) shows a decrease in signal intensity as a B1 artifact predicted by the simulations (e.g., Figure 9a top arrow). In part, these artifacts could also be susceptibility artifacts, due to the different magnetic properties of the metals with respect to the human tissue [60].”
- Related to the trace resistance, the following can be found: “The SD of trace resistance for the thin-film fabrication of traces was 0.49kOhm. Whereas a previous prototype was manufactured using Polymer Thick Film (PTF) with a similar technology [35], the resistance was found to be, on average 20.49 kOhm, with a larger SD of 0.85 kOhm for 256 traces.”, “The thin-film traces were fabricated, and the resistance was found to be, on average 12.23 kOhm, with a minimum Standard Deviation (SD) of 0.49 kOhm, with a target resistance between 10 kOhm and 15 kOhm (Figure 4).” and so on. In this regard, it is clear that the trace resistance should be between 10 kOhm and 15 kOhm. However, it is not explained why. What would happen if this resistance is higher or lower than the target resistance values?
Re: The discussion now explains better the reason behind the target values that were chosen for the NeoNet:
“In this study, our target trace resistance of 10-15kW was based on our recent study [55], in which we studied the safety using resistors of 10kW, and the 5-15kW industry standard. A lower resistance below 10kW, may affect the amount of RF-induced currents in the NeoNet traces inside a 3 T MRI, which could lead to both heating and greater B1 artifacts. For very total high resistances of the trace resistance and the contact EEG resistance on the subject, the EEG quality may be compromised and may damage the amplifier (maximum total resistance allowed may vary depending on the manufacturer).”
- . Sub-section 5.2 has the form of a literature review, not the form of any discussion of the obtained results.
Re: Section 5.2 was re-written to cite the results from the simulations in Sim4life and now the text reads:
“5.2 MRI Safety.
The use of conductive EEG leads in the presence of a radio-frequency field generates induced currents on the leads [37, 38] and potential increases of RF power absorbed in the human head, specified regarding Specific Absorption Rate (SAR) [39]. For relatively high-power RF sources, such as MRI RF coils, such interactions pose serious thermal-related safety risks of tissue heating and burns [40-45]. Finite Difference Time Domain [46] numerical estimations using a high-resolution model of head tissues [47] with EEG leads during MRI scanning simulation [37, 48, 49] showed the presence of local SAR peaks near EEG electrodes, with values depending on a wide range of variables, namely lead orientation shielding effect of EEG leads [50], RF frequency [49, 50], number of EEG electrodes/leads [49], and resistivity of EEG leads [10, 39]. These simulations, followed by temperature measurements during MR scanning with an anthropomorphic phantom, are essential for establishing system safety. Numerical simulations and temperature measurements conducted in such studies indicated that the use of metallic leads during MRI should be avoided [49] and that resistors (currently the 5-15kW are the industry standard) placed on the leads do not offer reliable protection against the antenna effect of the leads exposed to the MRI RF-field [51, 52]. In contrast, the increased lead resistivity afforded by PTF/thin-film technology allows for safe and high-quality recordings up to 7 T [53]. In regards to EEG-fMRI safety in infants/toddlers, we could only find a single study [54], which found that on a phantom, an MRI sequence (T2 with Max Turbo factor 25; SAR 89%) caused a large temperature increase at one electrode (Fpz; +4.1 C). Also, this study concluded, "Based on our findings, we strongly recommend against using the structural images obtained during simultaneous EEG-MRI recordings for further anatomical data analysis ". Our previous studies showed the MRI RF safety of the NeoNet using numerical simulation [19]. This study shows that the high-power MPRAGE heating was well below 2°C. Our recent simulation study on EEG safety with an infant/toddler [55] also showed a similar finding, based on the dielectric properties of the resistive traces in infants/toddlers for safe EEG-MRI [56], that thin-film resistive traces have reduced the RF heating at 3 T MRI within the safety guideline by FDA [57]. Thus, resistive traces may improve the safety of infants/toddlers EEG-MRI as shown by our SAR and thermal safety [10, 58], regardless of the sequence used. In this study, our target trace resistance of 10-15kW was based on our recent study [55], in which we studied the safety using resistors of 10kW, and the 5-15kW industry standard. A lower resistance below 10kW, may affect the amount of RF-induced currents in the NeoNet traces inside a 3 T MRI, which could lead to both heating and greater B1 artifacts. For very total high resistances of the trace resistance and the contact EEG resistance on the subject, the EEG quality maybe compromised and may damage the amplifier (maximum total resistance allowed may vary depending on the manufacturer).”
- In the following “All images were appropriate for clinical reading as evaluated by the two neuroradiologists, and image quality was not affected by the application of the NeoNet EEG during the recording.”, the two neuroradiologists found that the image quality was not affected by the application of the NeoNet system. This is far away from the standardized scientific approaches.
Re: We agree with the reviewer and we have introduced a Table and modified the text after statistical processing of the Likert scale results:
|
|
Scan |
NeoNet EEG |
Reviewer 1 |
Reviewer 2 |
|
Subject 1 |
T1 |
No |
4 |
4.5 |
|
T1 |
Yes |
4 |
4.5 |
|
|
Subject 2 |
T1 |
No |
5 |
5 |
|
DTI |
No |
4 |
5 |
|
|
T1 |
Yes |
5 |
5 |
|
|
DTI |
Yes |
4 |
5 |
|
|
Subject 3 |
T1 |
No |
5 |
5 |
|
T2 |
No |
4 |
4.5 |
|
|
T1 |
Yes |
5 |
5 |
|
|
T2 |
Yes |
4 |
4 |
The text was eliminated and substituted with:
“The MR images with NeoNet and without, both, enabled the pediatric neuroradiologist (P.E.G.) to clinically identify the left parietal atrophy in subject 2 indicated by the widened subarachnoid space with larger sulci and thinner gyri of the left parietal area of the brain (Figure 7 arrows on MPRAGE Subject 2) and the less prominent white matter tracts on the left side compared to the right side (Figure 7 circles on DTI Subject2). Additionally, in Subject 3, P.E.G. also identified the left sided tubers characterized by loss of the grey-white matter margin and somewhat irregular suggestive of some potential areas of subtle polymicrogyria (Figure 7 MPRAGE and T2). Furthermore, when two neuroradiologists (M.H.L. and R.G.) performed a double blinded quantitative asessment on the MR images using the Likert scale, both the NeoNet and the NoNet scored (4.8, 1) with a p-value of 0.34, indicating no statistical difference between the two types of MR images (Table 4).. However, the absence of CT artifacts with the NeoNet was evident in all slices due to the low-density materials (e.g., aluminum) used in the NeoNet that avoided streak artifacts.“
|
c |
|
Figure 7: MRI data with NoNet (left) vs. NeoNet (right). Both the NoNet (left) and NeoNet (right) enabled the pediatric neuroradiologist P.E.G. to clinically identify the lesions. Subject 2: Comparison of T1 Magnetization Prepared Rapid Gradient Echo (MPRAGE) and diffusion tensor imaging (DTI) sequences of a five years old female with left-side parietal atrophy (indicated by arrows on the MPRAGE and Circles on the DTI that point to the differences of the white matter tracts between the left and the right hemisphere). Subject 3: Comparison of MPRAGE, and T2 sequences of a five years old male with left side tuberous malformations (Arrow), shows the similar image quality of NoNet (left) and with NeoNet (right). |
- The conclusions do not have any scientific significance, and must be reorganized taking into account the scientific contributions, if any.
Re; The author recognize and has completely re-written the conclusions as follows:
“Metallic artifacts and distortions can substantially degrade the image quality of both MRI and CT when EEG electrodes are present. Thus, we have shown the novel thin-film trace EEG net ("NeoNet") results in improved MRI and CT image quality without compromising EEG signal quality. The 50cm long thin film trace was constructed from 30nm aluminum film, resulting in low density for reducing CT artifacts and 12 kW high-resistive low for reducing MRI artifacts. The NeoNet demonstrated safety in 3 Tesla (T) MRI for children with an increase of just 0.84 ËšC after 30 minutes of high-power scanning, which is the acceptance criteria for the temperature in 1 hour of normal operating mode scanning per the FDA. We investigated the effects of EEG nets on B1 transmit field distor-tion in 3T MRI with electromagnetic simulations, which predicted a 65% B1 transmit field distortion higher for commercially available copper-based EEG nets over the NeoNet.
No significant difference in the degree of MRI artifact or image distortion was found while analyzing the Likert scale responses from two board-certified neuroradiolo-gists blinded to the presence or absence NeoNet. Finally, NeoNet did not impact the quali-ty of EEG recording and allowed 128 dense EEG spatial sampling channels for clinical ep-ilepsy diagnostics.
The proposed NeoNet device, therefore, has the potential to allow concurrent EEG acquisition and MRI or CT scanning without significant image artifacts, facilitating clini-cal care and EEG/fMRI pediatric research. “
- What is the relevant state-of-the-art for the topic considered? Some of the references used in this manuscript are ancient.
Re: The Introduction now has a more extensive state-of-the-art description and in order to have more up to date references we replaced older references. Some older references are still in the manuscript because of historical reasons (like the early modeling of thin film) and because of biocompatibility studies are still valid and have not been (to the best of our knowledge) repeated since. The introduction now reads:
“This paper presents a novel high-density EEG net (i.e., NeoNet see Figure 1) using aluminum thin-film nanostructure for cloaking at Computerized tomography (CT) and MRI. The NeoNet is a step forward compared to the state-of-the-art polymer thick film nets [4] by providing more accurate control of trace electrical properties. Here we report a scalable approach to produce micro and nanoscale structures (Figure 2) of aluminum EEG traces with an ultra-high-aspect ratio (up to 17,000:1, with dimensions 30nm×50.8cm×100µm). Thus, the real challenge was constructing long, flexible, and narrow aluminum thin film traces for the NeoNet. To the best of our knowledge, this has not been reported yet. The major strengths of the NeoNet lie in the ease of application of the net compared to single electrodes in fragile patients such as neonates and children, and its compatibility with MRI environments, not needing to remove electrodes prior to the recording and re-apply them after the exam. Finally, other potential applications of the NeoNet also include electrical impedance spectroscopy (EIS) in children [3].
.”
- Figures 2 and 3 appear in the text after Figure 4. They should appear after their first mentions in the text.
Re: The order in which figures and Tables have been carefully revised.
- The Authors must find a way to include supplementary material in work, and only the material which has scientific significance.
Re: The Supplementary Material file that was uploaded with the first submission, has been trimmed down to the essential.
- Keywords should be listed in alphabetical order.
Re: The keywords are now in alphabetical order.
- There are some typos in this manuscript. For instance: “…applications Of the…”, “…artifacts., thus traces…”, etc.
Re: The typos have now been corrected and the entire text has been reviewed for spelling errors.
- Some abbreviations such as “GPU” or “TSE” are used without the associated descriptions.
Re: All the abbreviations and acronyms have now been associated with descriptions.
- The presentation style of this manuscript must be completely modified from the aspect of the scientific contribution.
We thank the reviewer for the comments and as pointed out in response N.2 we have worked very hard in adding new Figures, Tables and Data in this new version of the manuscript. We are confident of the main scientific contribution to the field of sensor science and technology of this new revision.

Round 2
Reviewer 2 Report
The Authors addressed adequately all comments of this reviewer. Congratulations.